# Data Release: DNA Barcodes of Plant Species Collected for the Global Genome Initiative for Gardens (GGI-Gardens) II

Morgan R. Gostel [1,2,*] , Mónica M. Carlsen [3] , Amanda Devine [2], Katharine B. Barker [2],
Jonathan A. Coddington [2] and Julia Steier [2]

1   Botanical Research Institute of Texas, Fort Worth, TX 76132, USA
2   National Museum of Natural History, Smithsonian Institution, P.O. Box 37012, Washington, DC 20013, USA;
    devinea@si.edu (A.D.); barkerk@si.edu (K.B.B.); coddington@si.edu (J.A.C.); steierj@si.edu (J.S.)
3   Missouri Botanical Garden, St. Louis, MO 63110, USA; monica.carlsen@mobot.org
*   Correspondence: mgostel@brit.org; Tel.: +1-(817)-463-4199

**Abstract:** The Global Genome Initiative for Gardens (GGI-Gardens) is an international partnership of botanic gardens and arboreta that aims to preserve and understand the genomic diversity of plants on Earth. GGI-Gardens has organized a collection program focused on the living collections that partner institutions and supports the preservation of herbarium and genomic vouchers. Collections made through GGI-Gardens are deposited in recognized herbaria and Global Genome Biodiversity Network-partnered biorepositories worldwide, meaning that they are made available to the public. With support from its parent organization, the Global Genome Initiative (GGI), plant DNA barcode sequencing is performed using tissues collected through this partnership that represent taxa without barcode sequences in GenBank. This is the second data release published by GGI-Gardens and constitutes 2722 barcode sequences from 174 families and 702 genera of land plants. All DNA barcodes generated in this study are now available through the Barcode of Life Data Systems (BOLD) and GenBank.

**Keywords:** biobanking; DNA barcoding; GenBank; ITS2; *matK*; *psbA-trnH*; *rbcL*; viridiplantae





## 1. Introduction

Founded in 2015, the Global Genome Initiative for Gardens (GGI-Gardens, [1]) is an international partnership of botanic gardens and arboreta that aims to preserve and understand Earth's genomic diversity of plants. GGI-Gardens supports the collection of both herbarium and genomic voucher material from the living collections in these partner gardens following best practices for herbarium and genomics research [2]. Collections made through this program are stored in Global Genome Biodiversity Network (GGBN)-partnered DNA banks [3], meaning that they can be utilized for applications ranging from whole genome sequencing [4] to DNA barcoding [5], as well as other genomic research.

Since their conception in 2003 [6], DNA barcode sequences have been used as powerful tools that enable the large-scale and rapid taxonomic identification of species for myriad purposes, including conservation [7], forensics [8], and the quantification of species diversity [9], among others. Emerging techniques, such as metabarcoding [10,11], leverage high-throughput sequencing technology and are capable of sequencing a mixed or pooled sample of species and identify them from their barcode sequence.

An important limiting factor for these and other studies that utilize DNA barcode sequences, however, is the representation of species diversity in reference databases [12]. DNA barcode reference databases are growing in both their taxonomic and geographic scope thanks to a number of large initiatives, which often focus on a particular branch of the tree of life or geographic area. For example, since 2005, the African Centre for DNA Barcoding has been contributing DNA barcode reference sequences from Africa to facilitate improved DNA barcoding applications from this continent [13]. The basic concept of DNA

barcoding has also expanded during recent years, thanks to high-throughput sequencing (HTS) technology and methods to "extend" the traditional barcode concept include the use of so-called "genome-skim" data [14] or whole organelle genome sequences as "super-" [15] or "ultra-barcodes" [16]. Clade-based approaches are contributing large-scale DNA barcode reference sequences for entire groups of organisms that are often regionally focused [17] or even hyper locally focused (e.g., sequencing living collections from botanic gardens, [18]), and in this paper we provide a large contribution from collections made by the GGI-Gardens program.

Facilitated by the Global Genome Initiative based at Smithsonian Institution (https://naturalhistory.si.edu/research/global-genome-initiative, accessed on 30 January 2022), new families and genera collected by GGI-Gardens partners to date have been extracted and sequenced using four plant DNA barcode loci (*rbcL*, *matK*, ITS2, and *psbA-trnH*). Past collections made through the GGI-Gardens program have been published as part of large DNA barcode "data releases", the first of which included the publication of nearly 2000 barcode sequences [5]. This manuscript represents the second data release for samples collected through the GGI-Gardens program and will serve as a significant contribution to available plant DNA barcode sequences in public repositories. These barcode sequences will facilitate future plant biodiversity research by improving the ability of researchers to use DNA barcode sequences to accurately identify species from DNA barcode reference databases through general plant inventories, ecological studies, and metabarcoding studies.

## 2. Methods and Materials

### 2.1. Tissue Collection

DNA barcode sequences published as part of this data release comprise collections conducted from two GGI-Gardens partners—the Botanical Research Institute of Texas (BRIT) and the Missouri Botanical Garden (MOBOT) between 2017 and 2020. A total of 817 collections (from 788 species) are represented in this data release, and these include 174 families and 702 genera (Supplementary Table S1). All collections were conducted following published best practices [2] and include herbarium vouchers, as well as genomic vouchers that include tissue preserved in silica gel and flash frozen in liquid nitrogen. Collections were prioritized using the GGI Gap Analysis Tool (https://globalgeno.me, accessed on 30 January 2022) following the scheme proposed in Linsky and Gostel [19] and whether DNA barcode quality sequences were available in GenBank. Silica-dried leaf tissues were sampled to generate DNA barcode sequences published in this study.

### 2.2. DNA Extraction

Silica-preserved leaf tissues (~10 µg of each sample) were sampled with 25 µL ETOH (to mitigate static) into a 96-well plate preloaded with glass and ceramic beads and then disrupted using a FastPrep 96 instrument (MP Biomedicals, Santa Ana, CA, USA). Whole genomic DNA was isolated using an AutoGenprep 965 (Autogen, Holliston, MA, USA) automatic extractor following the manufacturer's protocol for plant tissue.

### 2.3. PCR Amplification and Sequencing

Four standard plant DNA barcode loci (Table 1) were amplified, including the two core barcoding regions for land plants, *rbcL* and *matK* [20] and two additional loci that have been proposed as additional plant DNA barcoding loci, ITS2 and *psbA-trnH* [21–23]. PCR was performed using Bioline Taq polymerase (New England Biolabs, Ipswich, MA, USA) and a standard thermal cycling profile including an initial denaturation for 5 min at 95 °C, 35 cycles each including 95 °C denaturation for 30 s, a locus-specific annealing temperature (see Table 1) for 30 s, and an extension cycle at 72 °C for 40 s, followed by a final extension of 72 °C for 10 min. Amplified PCR products were cleaned using ExoSAP-IT (ThermoFisher Scientific, Waltham, MA, USA) following the manufacturer's protocols. Cycle sequencing was performed in 96-well plates using the same PCR primers, the BigDye® Terminator v3.1 Cycle Sequencing Kit (Applied Biosystems®, Norwalk, CT, USA), and

sequenced on the Automated ABI3730 Sequencer (Life Technologies, Carlsbad, CA, USA). Raw chromatograms were edited in Geneious Prime (2021, https://www.geneious.com, accessed on 15 January 2022) and annotated before uploading to BOLD (https://www.boldsystems.org, accessed on 30 January 2022) and GenBank (See Supplementary Table S1 for accession numbers).

**Table 1.** Locus information for each plant DNA barcoding marker used in this study, including forward and reverse primer names, primer sequences, annealing temperature, and citation.

| Locus | Primer Name | Forward Primer Sequence | Annealing Temperature | Citation |
|---|---|---|---|---|
| *rbcL* | rbcLa-F | ATGTCACCACAAACAGAGACTAAAGC | 55 °C | [24] |
| | rbcLa-R | GTAAAATCAAGTCCACCRCG | | [25] |
| *matK* | matK-xf | TAATTTACGATCAATTCATTC | 54 °C | [26] |
| | matK-MALP | ACAAGAAAGTCGAAGTAT | | [27] |
| ITS2 | ITS_S2F | ATGCGATACTTGGTGTGAAT | 56 °C | [28] |
| | ITS4 | TCCTCCGCTTATTGATATGC | | [29] |
| *psbA-trnH* | psbA3_f | GTTATGCATGAACGTAATGCTC | 64 °C | [30] |
| | trnHf_05 | CGCGCATGGTGGATTCACAATCC | | [31] |

## 3. Results and Data Resources

*Sequence Characteristics and Upload to BOLD and GenBank*

A total of 2722 DNA barcode sequences were generated and uploaded to the BOLD and GenBank reference databases (both publicly available, Supplementary Table S1). Most DNA barcode sequences represented the *rbcL* locus (789), and the fewest sequences were generated for the *matK* locus (597 sequences). Overall, 650 and 686 sequences were generated for ITS2 and *psbA-trnH*, respectively. Among the DNA barcode sequences presented in this data release are 12 families, 292 genera, and 604 species that previously did not have barcode sequence data available in GenBank. All sequences uploaded to BOLD are contained within the BOLD projects GRDTX and GRDMO. All sequences uploaded to GenBank are part of the GGI-Gardens Bio-Projects (ID: PRJNA791936 & PRJNA485943).

These sequences represent important resources for biodiversity studies and will facilitate rapid species identification and ecological studies that seek to understand plant community composition [9] and species interactions [32], as well as conservation assessments that depend upon DNA barcoding to identify and control invasive species, e.g., [33] and enforce policies regarding the trade in endangered plants, e.g., [34,35]. Expanding the plant DNA barcode sequence reference library can also help botanic gardens to accurately identify species in their living collections, i.e., [36]. We hope this work encourages others who work with plant DNA barcodes to contribute to growing DNA barcode reference databases to improve these public resources.

**Supplementary Materials:** The following supporting information can be downloaded at: https://www.mdpi.com/article/10.3390/d14040234/s1. Table S1. List of samples collected for the Global Genome Initiative for Gardens projects selected for DNA barcoding in this study, with GenBank accession numbers and genomic voucher identification numbers. All the sequences are included in the GGI-Gardens BioProjects PRJNA791936 and PRJNA485943 in GenBank and BOLD projects GRDTX and GRDMO. Each Collector Number includes a link to the digital genomic voucher record stored in the Smithsonian Biorepository.

**Author Contributions:** Conceptualization, M.R.G. and J.S.; methodology, M.R.G., M.M.C. & J.S.; software, A.D.; formal analysis, J.S.; resources, K.B.B. and J.A.C.; data curation, J.S.; writing—original draft preparation, M.R.G.; writing—review and editing, J.S., M.M.C., M.R.G., A.D., K.B.B., J.A.C.; project administration, M.R.G., K.B.B., J.A.C. All authors have read and agreed to the published version of the manuscript.

**Funding:** This research was funded by the Global Genome Initiative under grants GGI-Gardens-2019-206, GGI-Gardens-2020-246, and GGI-Partnerships-2017-167.

**Institutional Review Board Statement:** Not applicable.

**Data Availability Statement:** The data presented in this study are available in Supplementary Table S1.

**Acknowledgments:** All laboratory work was conducted in the Laboratories for Analytical Biology (LAB) at the National Museum of Natural History in Washington, DC and at the Museum Support Center in Suitland, MD. Genomic vouchers are deposited in the Smithsonian Institution Biorepository as well as the DNA banks at the Botanical Research Institute of Texas (BRIT) and the Missouri Botanical Garden (MO). Herbarium vouchers for samples associated with Genbank Projects PRJNA791936 and PRJNA485943 have been deposited in the BRIT and MO herbaria, respectively. M.R.G. thanks Faranhoz Khojayori, Seth Hamby, and Jerrod Stone for their assistance with the collection of samples used in this work from BRIT. M.M.C. thanks Danielle Hopkins, Stephanie Keil, Ella Ludwig, and Gabrielle McAuley for their assistance with collections made at the Missouri Botanical Garden. The sequences published as part of thisfFigur work were supported by the Global Genome Initiative, with assistance from Jose Zúñiga. Any paper(s) resulting directly from this specimen-processing project should reference support from the Global Genome Initiative and the Laboratories of Analytical Biology, National Museum of Natural History, Smithsonian Institution.

**Conflicts of Interest:** The authors declare no conflict of interest.

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
