# Peer review of "Data Release: DNA Barcodes of Plant Species Collected for the Global Genome Initiative for Gardens (GGI-Gardens) II"

_diversity, doi:10.3390/d14040234_

Round 1

Reviewer 1 Report

Dear Authors, I am happy to see this manuscript being almost ready. This is a great contribution to the global plant DNA barcode reference library for four most widely used markers.

  • However, I think it would be appropriate to discuss why the decision was made to concentrate the efforts on the traditional DNA barcodes, not on the shotgun sequencing.
  • An interesting aspect, but not discussed in this paper:

https://doi.org/10.1371/journal.pone.0235569

Line 23. Adding the markers to the key words helps a lot to screen the paper, and to find necessary information.

Line 48. I found irrelevant here citation of two publications about DNA barcoding of fish and insects as the examples of the local DNA barcode libraries. It would be appropriate if there were no other publications exemplifying the regional plant DNA barcoding datasets. However, we have enough of those examples these days. I hope you will forgive me my self-citation, but the DNA barcoding of the vascular plants of Canada might be more appropriate here…:). The following publications may be helpful to discuss the first bullet point:

10.1101/2021.08.11.456029

https://doi.org/10.1093/gigascience/giz007

Line 53. Please use Italic for the names of the following markers across all manuscript and supplemental material: rbcL, matK, psbA-trnH. Based on the information provided in the Table 1 you sequenced ITS2 region, which is the second part of the internal transcribed spacer ITS. Please correct it across all manuscript.

Line 53-56. Style: I think the sentence should be slightly modified to be easier readable.

Line 65. Style: plural and single forms for the same noun in one sentence (voucher). I suggest them all plural.

Line 98. The word “successfully” is redundant.

Author Response

Dear Authors, I am happy to see this manuscript being almost ready. This is a great contribution to the global plant DNA barcode reference library for four most widely used markers.

However, I think it would be appropriate to discuss why the decision was made to concentrate the efforts on the traditional DNA barcodes, not on the shotgun sequencing.

An interesting aspect, but not discussed in this paper: https://doi.org/10.1371/journal.pone.0235569

Good question, we have added this citation as well as two others that highlight the use of high-throughput sequencing approaches for DNA barcoding and encourage the use of collections made through this project for larger-scale genomic projects (including shotgun sequencing) on lines 46–53.

Line 23. Adding the markers to the key words helps a lot to screen the paper, and to find necessary information.

Great suggestion, thank you – we have added these to the Key Words.

Line 48. I found irrelevant here citation of two publications about DNA barcoding of fish and insects as the examples of the local DNA barcode libraries. It would be appropriate if there were no other publications exemplifying the regional plant DNA barcoding datasets. However, we have enough of those examples these days. I hope you will forgive me my self-citation, but the DNA barcoding of the vascular plants of Canada might be more appropriate here…:). The following publications may be helpful to discuss the first bullet point: 10.1101/2021.08.11.456029 & https://doi.org/10.1093/gigascience/giz007

We very much agree with this point – both citations you reference are relevant replacements for the previous citations (Rashman et al. 2019 and Ferreira et al. 2020). We have replaced both of these with the first reference (Chua et al. 2021) as a great example of regional barcoding of the Danish flora. This also helps reflect the trend toward the “extended barcode” that was raised in your first comment and we hope encourages other researchers to move toward more standardized approaches to the “extended barcode”. The second citation has also been added (as a second sentence) that highlights localized DNA sequencing; in this from living collections in a botanic garden. We (the authors) were not aware of this publication – it’s an excellent paper and very relevant to our article here.

Line 53. Please use Italic for the names of the following markers across all manuscript and supplemental material: rbcL, matK, psbA-trnH. Based on the information provided in the Table 1 you sequenced ITS2 region, which is the second part of the internal transcribed spacer ITS. Please correct it across all manuscript.

Thank you – we have formatted the names of each gene region barcode marker in italics; we have also revised the name of ITS as ITS2.

Line 53-56. Style: I think the sentence should be slightly modified to be easier readable.

We agree, this sentence has been revised for clarity.

Line 65. Style: plural and single forms for the same noun in one sentence (voucher). I suggest them all plural.

Thank you, we have changed as suggested (made the first use of “voucher” plural).

Line 98. The word “successfully” is redundant.

Thank you, this has been removed for clarity.

Reviewer 2 Report

The DNA barcoding is a valuable methodology to establish the genetic data bank for conservation of biodiversity.

However, the findings of this study are not described in terms of biodiversity

The standard structure of introduction (e.g. hypothesis, necessity, aims and etc.) should be considered exactly.

The aims should be described clearly.

Where is the supp. table?

What is the main achievements the study in biodiversity as well conservation approach?

Discussion is very brief .

Please clearly describe applied achievements

Author Response

The standard structure of introduction (e.g. hypothesis, necessity, aims and etc.) should be considered exactly.

When writing this article, we paid close attention to the instructions for authors with regard to this section: “The introduction should briefly place the study in a broad context and highlight why it is important. It should define the purpose of the work and its significance, including specific hypotheses being tested. The current state of the research field should be reviewed carefully and key publications cited. Please highlight controversial and diverging hypotheses when necessary. Finally, briefly mention the main aim of the work and highlight the main conclusions. Keep the introduction comprehensible to scientists working outside the topic of the paper.”

We understand that the necessity and aims can be improved and we have added some introductory information (also consistent with suggestions from Reviewer 1) to improve the scope and context that we hope better illustrate the necessity and aims of this work. See lines 46–53 and lines 62–66.

The aims should be described clearly.

The core aim of this study was to expand the representation of species diversity across the plant tree of life in current plant DNA barcoding databases. We mentioned this in the second paragraph of the introduction, but this reviewer makes a good point that the aims should be described more clearly and we have elaborated this point for readers on lines 62–66.

Where is the supp. table?

This was uploaded as a .zip file (per submission instructions) – was this not available to reviewers?

What is the main achievements the study in biodiversity as well conservation approach?

The main achievements from this study that have implications for biodiversity and conservation research is the availability of nearly 3,000 newly published plant DNA barcode sequences that have been deposited in publicly available reference databases. We have added text to clarify this on lines 144 and lines 152–160.

Discussion is very brief .

We have added a new paragraph that places the discussion of results in the broader context of research using plant DNA barcodes for ecology, evolution, and conservation.

Please clearly describe applied achievements

As with the reviewer comment above, we have highlighted several applications of the plant DNA barcode sequences that result from this work in lines 152–160; notably that these sequences serve as permanent resources deposited in reference databases that will be used for subsequent biodiversity and conservation studies.